# Impaired Well-Being and Insomnia as Residuals of Resolved Medical Conditions: Survey in the Italian Population

**DOI:** 10.3390/ijerph21020129

**Published:** 2024-01-24

**Authors:** Danilo Menicucci, Luca Bastiani, Eleonora Malloggi, Francesca Denoth, Angelo Gemignani, Sabrina Molinaro

**Affiliations:** 1Department of Surgical, Medical and Molecular Pathology and Critical Care Medicine, University of Pisa, 56126 Pisa, Italy; danilo.menicucci@unipi.it (D.M.); eleonora.malloggi@med.unipi.it (E.M.); angelo.gemignani@unipi.it (A.G.); 2Institute of Clinical Physiology, National Research Council, 56124 Pisa, Italy; lucabastiani@ifc.cnr.it (L.B.); francesca.denoth@ifc.cnr.it (F.D.); 3Clinical Psychology Branch, Azienda Ospedaliero-Universitaria Pisana, 56100 Pisa, Italy

**Keywords:** sleep, chronic inflammation, psychiatric disorders, anxiety, depression, clinical disorders

## Abstract

Background: Well-being encompasses physical, mental, social, and cultural aspects. Sleep quality and pathologies are among the objective conditions jeopardising it. Chronic insomnia, inflammatory-based diseases, and mood disorders often occur in a single cluster, and inflammation can negatively impact sleep, potentially harming well-being. Some evidence from specific clinical populations suggests that also some resolved past diseases could still have an impact on present sleep quality and well-being. The aim of the present study is to investigate, in the general population, whether and to what degree well-being and insomnia are associated with resolved pathologies. Methods: A cross-sectional survey (IPSAD^®^) was carried out using anonymous postal questionnaires that investigated past and present general health, well-being, and insomnia. A total of 10,467 subjects answered the questionnaire. Results: Several classes of both current and resolved pathologies resulted in increased odds ratios for current insomnia (odds ratios = 1.90; 1.43, respectively) and impaired well-being (odds ratios = 1.75; 1.33, respectively), proportional to the number of the displayed pathologies. Notably, both current and resolved past psychiatric disorders were strongly associated with both current impaired well-being (odds ratios = 5.38; 1.70, respectively) and insomnia (odds ratios = 4.99; 2.15, respectively). Conclusions: To explain these associations, we suggest that systemic inflammation conveyed by several medical conditions disrupts homeostatic processes, with final effects on sleep quality and behaviour.

## 1. Introduction

According to the World Health Organization (WHO), “Well-being is a positive state experienced by individuals and societies. Similar to health, it is a resource for daily life and is determined by social, economic and environmental conditions. Well-being encompasses quality of life and the ability of people and societies to contribute to the world with a sense of meaning and purpose” [1]. Good well-being is one of the primary aims of human beings; however, its maintenance is a difficult due to its multifactorial nature. Indeed, it is influenced by reciprocally connected physical, mental, social and cultural aspects. Besides the subjective factors affecting well-being [2], there are some objective conditions that jeopardise it: among them are being affected by pathologies and a poor quality of sleep, which has a tangible influence on individuals’ daily functioning [3,4,5].

Some medical conditions are very frequent among European populations: more than 80 million people are affected by cardiovascular diseases, which represent 45% of all causes of death [6]. Also, more than 52 million people suffer from diabetes, and more than 2.7 million people are diagnosed each year with cancer, which nowadays represents the second cause of mortality in Europe. Neurological and psychiatric disorders are the third cause of premature mortality; indeed, 83 million people suffer from a psychiatric disorder, 40% of which is represented by depression [7].

Another condition that is widely spread in Western society is insomnia. More than one half of adults have been reported to display sleep difficulties and, among them, 22% meet the diagnostic criteria for insomnia. The most prevalent symptom is difficulty in maintaining sleep (61%), followed by nonrestorative sleep (25.2%), difficulty initiating sleep (7.7%), and early morning awakening (2.2%). Insomnia has a higher prevalence in women, in the elderly, in individuals with a low socioeconomic status, and in those with poor health or low levels of well-being [8]. In turn, insomnia can seriously affect well-being: indeed, one of the underestimated insomnia-related factors is its impact on daily life, including daytime sleepiness [9]. In particular, a recent study found that, in Italy, the prevalence of insomnia among adult people (≥50 years old) is more than 55% [10].

A significant number of clinical and epidemiological studies have highlighted that insomnia and medical conditions are exacerbated by each other in a vicious cycle [11,12,13,14,15,16,17], thus producing negative consequences on well-being [18]. In this context, it has been reported that chronic insomnia acts as a predisposing, precipitating and perpetuating factor for several medical conditions. In particular, Hypothalamic–Pituitary–Adrenal (HPA) axis hyperactivation, sympathetic hyperactivation, higher levels of cortisol and proinflammatory cytokines induced by insomnia [19] seem to lay the foundation for the onset of cardiovascular [20,21] and autoimmune diseases [22], and finally for impairing daily functioning and one’s general quality of life. Furthermore, chronic insomnia seems to increase the risk of breast, prostate, colon, endometrial epithelial malignancies, and non-Hodgkin’s lymphoma [23,24]. Biomolecular studies have corroborated this evidence, reporting that circadian disruption induces the epigenetic modification of circadian genes, which serve as transcriptional regulators that affect the expression of many cancer-related genes [25,26]. Also, according to epidemiological studies, about 50% of people diagnosed with insomnia meet the diagnostic criteria for a psychiatric disorder, especially anxiety or mood disorders [8,12]. In fact, the disruption of cortisol negative feedback [27] and increased excitotoxicity and neuronal apoptosis [28] have been observed in both chronic insomnia and depression; meanwhile, cognitive and psychophysiological hyperarousal with beta and gamma activity intrusions in NREM (Non-Rapid Eye Movement) sleep have been reported in insomnia [29,30] and in anxiety disorders [31,32].

On the other hand, chronic insomnia has been shown to be induced by several medical conditions; among them, inflammatory and autoimmune diseases have been reported to cause insomnia [33,34,35,36,37,38]. Indeed, the etiopathogenetic mechanism conveyed by inflammatory diseases comprises the dysregulation of the autonomic system and HPA axis, as well as alterations in immunological biomarkers; these factors can play a pivotal role in the onset of insomnia [39].

Despite the extensive body of literature supporting that insomnia is associated with several other medical conditions, few studies have investigated whether insomnia is reversed once the medical condition that may had favoured its onset has been resolved or whether, on the contrary, there are maintenance factors that can perpetuate insomnia, regardless of the disease resolution. In this context, only a few prospective studies have reported the development of chronic insomnia, and thus an impaired quality of life. These studies were performed on selected patients, sharing the same pathological condition, who underwent follow-up examinations after the resolution of a medical condition. One study revealed high levels of fatigue and insomnia in cancer survivors several years after diagnosis, even in the case of successful treatment [40]. These findings were further corroborated by another study on cancer survivors, which found that the patients displayed fatigue and insomnia symptoms four years after diagnosis; moreover, fatigue and insomnia were associated with high levels of C-Reactive Protein, one of the main inflammatory biomarkers, suggesting that the long-term pro-inflammatory cytokine cascade elicited by the disease might remain elevated over time and chronically disrupt sleep [40,41].

The aforementioned studies are the basis for the existence of putative mechanisms induced by inflammatory-based diseases that are perpetuated also after the medical resolution, and that could affect patients’ well-being and sleep. In order to test this hypothesis, further investigations including broader sample sizes and different types of pathologies are needed. Accordingly, the main aim of the present study is to investigate, by means of an epidemiological approach, whether well-being and insomnia are associated with resolved inflammatory-based pathologies in the general population.

## 2. Materials and Methods

Data for this study were drawn from the Italian Population Survey on Alcohol and other Drugs (IPSAD^®^) survey. This survey closely follows the guidelines provided by EMCDDA (European Monitoring Centre for Drugs and Drug Addiction) in Lisbon. A more detailed description of the IPSAD^®^ methodology is published elsewhere [42]. Over the course of the different waves, the questionnaire was enriched with forms on socio-environmental characteristics such as lifestyle, eating attitudes, metabolic syndrome-related diseases, psychological stress, depression, and gambling habits. Current data are related to the 2017–2018 edition of the survey, in which 10,467 individuals answered the questionnaire (https://www.epid.ifc.cnr.it/project/ipsad/ (accessed on 15 January 2024). The Cronbach’s alpha calculated from the data of this edition was greater than 0.8.

The assessment of perceived well-being was conducted by including the World Health Organization’s well-being scale (WHO-5) [43] in the questionnaire (General Health Section, item D1). WHO-5 is among the most widely used questionnaires for the assessment of subjective well-being; it provides a 0 to 100 total score and, according to a binary classification, scores below 50 suggest impaired well-being [44,45]. It was validated for both adolescents (14–19 years) and adults (>19 years) [44].

Insomnia was assessed according to the Insomnia Severity Index (ISI) scale [46], which was included in the General Health Section of the questionnaire. The Insomnia Severity Index (ISI) [47] is a brief self-report instrument measuring the patient’s perception of his/her insomnia with regard to the prior 2 weeks. The instrument investigates difficulties regarding the onset and maintenance of sleep, satisfaction with one’s own sleep and its putative interference with daily functioning. The scale provides a total score ranging from 0 to 28, with higher scores indicating a greater severity of insomnia. Different thresholds have been proposed to classify insomnia according to ISI scores. According to a binary classification, scores of 10 or higher have been considered suggestive of the presence of insomnia symptoms in the general population [48], whereas introducing multiple thresholds led to a differentiation between the absence of insomnia (0–7); sub-threshold insomnia (8–14); and overt (moderate and severe) insomnia (15–28) [49]. It was validated for both adolescents (14–19 years) and adults (>19 years) [50].

### 2.1. Presence of Medical Conditions

Item D14 of the General Health Section in the IPSAD^®^ questionnaire was dedicated to acquiring information on current and past general health. The question aimed to investigate the presence of some classes of frequent diseases: diabetes; hypertension, myocardial infarction, angina or any ischemic cardiovascular diseases; stroke or other cerebrovascular diseases; other heart diseases (arrhythmias, heart valve diseases, heart infection, congenital diseases); high cholesterol; thyroid disorders; tumours (including lymphoma or leukaemia); psychiatric disorders (including anxiety, depression); joint and rheumatic pain (including fibromyalgia, rheumatoid arthritis etc.); diseases of the skin (psoriasis, eczema etc.); sleep apnoea; and cognitive impairment (memory loss, Parkinson’s disease, senile dementia etc.). Participants were asked “Has your doctor ever told you that you suffer from one or more of the following diseases/health conditions?”. For each disease class, participants had to mark all the appropriate options according to three response categories: “No”, “I suffered in the past (to be considered a period that is clearly separated from the present one)” and “I suffer now”. For the complete item, see Appendix A.

In the whole study, subjects who exhibited the same disease both currently and in the past were excluded from the analysis so that there was a sharp distinction between the medical conditions displayed only in the present and those displayed only in the past.

### 2.2. Statistical Analysis

In order to perform an analysis of the association between the WHO-5/ISI classification and the presence of both current and resolved medical conditions, binary or multinomial logistic models were estimated depending on the binary/multinomial classification of the scales.

The first models did not distinguish between disease classes, and they considered two independent variables: (1) *only present*, a binary variable for the presence of at least one current (and not pre-existent) disease, and (2) *only past*, a binary variable for the presence of at least one past but resolved disease.

Afterwards, instead of considering only the presence of diseases, we considered the number of displayed pathologies. In the same models, we therefore considered four levels for both *only present* and *only past* variables. The levels corresponded to the number of current and resolved diseases, respectively. The levels were as follows: none, only one, two, and three or more diseases.

Finally, we assessed the association between the WHO-5/ISI binary classification and each pathology by using binary logistic models. For every class of pathologies, two independent variables were taken into account: (1) *only present*, the binary variable indicating the existence of the pathology only in the present (not pre-existent), and (2) *only past*, the binary variable indicating the existence of the pathology only in the past. For this latter analysis, classes of pathologies that were displayed by less than one hundred individuals in the past or in the present were ruled out from the analyses in order to prevent the outcomes from being biased by an exiguous sample. Therefore, only hypertension, non-ischemic heart diseases, high cholesterol, thyroid disorders, psychiatric disorders, rheumatic pains, skin disorders and sleep apnoea were included in the analyses. The number of subjects displaying each disease is reported in paragraph 3.4 of the Results section.

For all the binary and multinomial logistic regression models, we adjusted the analyses by age and gender and their interaction. Other possible demographic variables (such as those in Table 1) were not considered in the models. The results are reported as an odds ratio (OR) with a 95% confidence interval (CI) and *p* values.

The statistical analysis was performed by employing Statistical Package for Social Science software (IBM Corp. Released 2021. IBM SPSS Statistics for Windows, Version 28.0., IBM Corp., Armonk, NY, USA) [51]. All *p* values were two-sided and considered statistically significant when below 0.05.

## 3. Results

Herein, we provide an overview of the main results, moving from the descriptive statistics of insomnia and well-being to their association with pre-existent and current pathologies. Medical conditions that occurred only in the present and only in the past were taken into account, in order to discern between the effects of the current pathologies and the effects of pathologies that were exhibited in the past and remedied. Table 1 shows the demographic characteristics of the study sample. Table 2 shows the number of males and females who displayed WHO-5 and ISI scores below or above the cut-off value and those who displayed one, two, three or no current and past pathologies.

Table 3 shows the number of participants currently displaying no pathologies or at least one pathology, and the mean scores of well-being and insomnia in relation to the two groups, respectively. The WHO-5 score of participants with no current pathologies was significantly different from the score of participants with at least one current pathology (t = −12.33; *p* < 0.0001), as well as the ISI scores (t = 10.89; *p* < 0.0001).

Table 4 shows the number of participants who displayed no pathologies or at least one pathology in the past, and the mean scores of well-being and insomnia in relation to the two groups, respectively. The WHO-5 score of participants with no past pathologies was significantly different from the score of participants with at least one resolved pathology (t = −6.43; *p* < 0.0001), as well as the ISI scores (t = 7.77; *p* < 0.0001).

### 3.1. Prevalence of Insomnia and Low Level of Well-Being and the Association between Them

In total, 10,467 subjects were recruited for the IPSAD^®^ survey.

Regarding well-being and insomnia, 5561 subjects fully completed both the WHO-5 and ISI surveys (for the aim of this study, response rate = 53.13%). Among them, 2416 displayed a score below the WHO-5 clinical threshold (≤50), namely 43.4% of respondents (women = 47.7%), and 1256, namely 22.6% (women = 25.7%), displayed an ISI score beyond the cut-off value (≥10), indicating the presence of insomnia symptoms. Appendix A show the associations between well-being and gender (Pearson’s X^2^ = 44.710; *p* = 0) and insomnia and gender (Pearson’s X^2^ = 32.941; *p* = 0), respectively.

In total, 72.1% of the subjects who displayed a score beyond 10 in the ISI also reported a score below 50 in the WHO-5, while 64.9% of subjects who reported a score below the clinical threshold also had impaired well-being (Pearson’s X^2^ = 540.45, *p* < 0.0001). For further details, see Appendix A.

The ISI scores were then stratified into three categories based on the insomnia severity level (score range of 0–7 for not clinically relevant insomnia symptoms; score range of 8–14 for subthreshold insomnia; score of ≥15 for moderate and severe insomnia). Among all the 5879 respondents, 28% displayed subthreshold insomnia, 56.1% of which were women (30.4% of all women respondents), while 6.1% showed moderate insomnia, 62.5% of which were women (7.4% of all women respondents). Considering the group displaying a score below the clinical threshold of WHO-5, 31.3% of subjects reported not having insomnia symptoms, 63.2% reported subthreshold insomnia, and 85.2% reported moderate insomnia (Pearson’s X^2^ = 698.814, *p* < 0.0001). For further details, see Appendix A.

### 3.2. Well-Being and Current and Resolved Medical Conditions Are Positively Associated

Table 5 shows the results of logistic regression analyses evaluating how well-being is associated with age, gender, the interaction between them, and with the presence of at least one pathology (ongoing or resolved). The analyses showed a strongly significant association between WHO-5 and the presence of current pathologies (OR = 1.75, *p* < 0.0001) and pathologies in the past (OR = 1.33, *p* < 0.0001). Neither age nor gender nor age × gender interaction effects were observed. For further details, see Appendix A.

Moving deeper into the study of the relationship between pathologies and well-being, the number of pathologies displayed in the present or in the past was considered. In total, 7231 subjects were included in this analysis, as only 69.1% of the sample fully completed both the WHO-5 and “General Health” sections.

As shown in Figure 1A, the relationship between low levels of perceived well-being and the presence of diseases strengthened parametrically with the increase in the number of pathologies displayed, and this association was stronger for the subgroup displaying three pathologies in the present. The association was significant with regard to the pathologies displayed in the present (OR = 1.44, *p* < 0.0001; OR = 1.91, *p* < 0.0001; OR = 3.45, *p* < 0.0001 for the presence of one, two or three pathologies, respectively); this also held true for the pathologies that had been resolved in the past (OR = 1.30, *p* < 0.0001; OR = 1.36, *p* < 0.0001; OR = 1.78, *p* < 0.0001 for the presence of one, two and three pathologies, respectively). For further details, see Appendix A. In summary, the presence of pathologies in the past and in the present negatively affected well-being in proportion to the number of diseases; this result was not affected by gender, age, and the interaction between them.

### 3.3. Insomnia and Current as Well as Resolved Medical Conditions Are Positively Associated

The results from the logistic regression analyses investigating the relationship between the ISI scores, demographic variables and pre-existent and current pathologies were also considered.

In total, 4835 subjects were included in this analysis, as only 46.2% of the subjects fully completed both the ISI and “General Health” sections.

Table 6 shows the results of the logistic regression analyses evaluating how insomnia is associated with age, gender, the interaction between them, and with the presence of at least one pathology (ongoing or resolved). The gender factor did not significantly alter the prevalence of insomnia, but age resulted in a slight reduction in the prevalence of insomnia (OR = 0.98, *p* = 0.017); this trend is apparent in middle-aged to older people (see Appendix A). The age × gender interaction was also shown to significantly influence the prevalence of insomnia (OR = 1.01, *p* = 0.010). Finally, exhibiting at least one pathology both in the present and in the past negatively affected sleep (OR = 1.90, *p* < 0.0001; OR = 1.43, *p* < 0.0001, respectively).

The relationship between insomnia and the presence of pathologies gradually strengthened with the increase in the number of pathologies exhibited, as shown by the odds ratio values in Figure 1B. This association was significant with regard to the pathologies that were displayed in the present (OR = 1.51, *p* < 0.0001; OR = 2.17, *p* < 0.0001; OR = 3.61, *p* < 0.0001 for the presence of one, two or three pathologies, respectively); this also held true with regard to the pathologies that were resolved in the past (OR = 1.37, *p* < 0.0001; OR = 1.59, *p* < 0.0001; OR = 1.73, *p* < 0.0001 for the presence of one, two and three pathologies, respectively). For further details, see Appendix A.

Concerning the relationship between the severity of insomnia and current and resolved medical conditions, regression models took into account the three levels of categorization of ISI scores, and coherent results were obtained.

A multinomial logistic regression analysis showed that displaying at least one current or past pathology was significantly associated with subthreshold insomnia (OR = 1.56, *p* < 0.0001; OR = 1.45, *p* < 0.0001, respectively). Moreover, when displaying at least one current pathology, the odds ratio doubled for moderate insomnia (OR = 3.08, *p* < 0.0001) compared to the probability of having subthreshold insomnia. Moderate insomnia was also significantly associated with the presence of at least one pathology in the past (OR = 1.82, *p* < 0.0001). Finally, multinomial logistic regression analyses revealed that the severity of insomnia gradually increases in proportion to the number of diseases displayed in the present or in the past. In this case, the age effect was significant regarding both subthreshold (*p* = 0.02) and moderate insomnia (*p* = 0.01). For further details, see Appendix A.

### 3.4. Association between Each Medical Condition and Insomnia as Well as Low Level of Well-Being

Finally, binary logistic models were used to investigate the effect of the presence of each pathology displayed only in the present or only in the past on the prevalence of insomnia and the well-being level. Figure 2A,B shows the influence of specific current or past pathologies on well-being and insomnia, respectively.

Among all the subjects who completed the survey, 1104 displayed hypertension in the present, and this had a significant influence on insomnia (OR = 1.33, *p* < 0.0001) and on well-being (OR = 1.44, *p* < 0.0001); 458 individuals reported that they had suffered from hypertension in the past, which, however, did not significantly influence either insomnia symptoms or well-being in the present. A significant age × gender interaction effect was observed for insomnia (*p* = 0.003).

In total, 330 individuals reported that they suffered from non-ischemic heart diseases (i.e., any congenital heart diseases, heart valve diseases, and arrhythmias) in the present, while 254 reported that they had suffered from these conditions in the past and had recovered. Both present and past non-ischemic heart disease was observed to have a significant impact on insomnia (OR = 1.572, *p* = 0.001; OR = 1.61, *p* = 0.004, respectively), while only non-ischemic heart disease in the present had a significant impact on well-being (OR = 1.34, *p* = 0.004). The age × gender interaction effect was significant for both insomnia (*p* = 0.011) and well-being (*p* = 0.007).

High levels of cholesterol were displayed by 1438 subjects in the present, and 728 had suffered from it in the past. Only a currently high cholesterol had a significant impact on insomnia (OR = 1.22, *p* = 0.022), while both current and past high cholesterol significantly affected well-being (OR = 1.33, *p* < 0.0001; OR = 1.23, *p* = 0.008, respectively). The age × gender interaction effect was significant for insomnia (*p* = 0.015).

In total, 551 subjects reported that they currently had thyroid disorders, while 312 reported that they had suffered from these conditions in the past. Only thyroid disorders in the present significantly influenced insomnia (OR = 1.46, *p* = 0.001), while both past and current thyroid disorders had a significant impact on present well-being (OR = 1.39, *p* < 0.0001; OR = 1.34, *p* = 0.010, respectively). The age × gender interaction effect was significant for both insomnia (*p* = 0.018) and well-being (*p* = 0.004).

In total, 924 subjects reported rheumatic pains in the present and 331 reported them in the past, but only the presence of rheumatic pains in the present had a significant impact on insomnia (OR = 2.13, *p* < 0.0001), while both past and current rheumatic pains significantly influenced well-being (OR = 2.05, *p* < 0.0001; OR = 1.57, *p* < 0.0001). The age × gender interaction effect was significant for both insomnia (*p* = 0.026) and well-being (*p* = 0.035).

In total, 666 subjects reported skin disorders in the present, which significantly influenced insomnia (OR = 1.61, *p* < 0.0001), while 722 suffered from skin disorders in the past. Both past and current skin disorders significantly affected present well-being (OR = 1.46, *p* < 0.0001; OR = 1.18, *p* = 0.027, respectively). The age × gender interaction effect was significant for both insomnia (*p* = 0.007) and well-being (*p* = 0.003).

In total, 333 subjects reported that they suffered from sleep apnoea in the present, while 232 reported that they had suffered from this condition in the past. Only present apnoea was reported to significantly affect sleep (OR = 2.57, *p* = 0), while both past and current sleep apnoea had a significant impact on well-being (OR = 1.86, *p* < 0.0001; OR = 1.415, *p* = 0.006, respectively). The age × gender interaction effect was significant for both insomnia (*p* = 0.001) and well-being (*p* = 0.006).

Finally, 720 subjects declared that they suffered from psychiatric disorders in the present, and 1075 reported that they had suffered from psychiatric disorders in the past. Both current and resolved psychiatric disorders had the strongest impact on insomnia symptoms (OR = 4.99, *p* < 0.0001; OR = 2.15, *p* < 0.0001, respectively) and well-being (OR = 5.38, *p* < 0.0001; OR = 1.70, *p* < 0.0001, respectively).

## 4. Discussion

The primary aim of the present study was to investigate, among subjects aged between 15 and 74 belonging to the Italian population, the association between resolved inflammatory-based pathologies (the number and type of pathologies) and the prevalence of insomnia, as well as individuals’ well-being level. The sharp distinction between ongoing and resolved pathologies precluded the inclusion of pathologies that had onset in the past and persisted in the present, thus preventing the assessment of their effect on sleep quality and well-being. Secondly, we investigated the association between insomnia and well-being. The investigation was conducted through the administration of a self-report questionnaire that was distributed via mail. Its cross-sectional design prevented the causal relationship between the aforementioned conditions being investigated; however, it was possible to clearly establish the association between insomnia, several types of pathologies and well-being in a large sample.

We found that all the medical conditions that were displayed in the present were strongly associated with both sleep quality and well-being, suggesting that disease-related inflammatory activity, immune system and neuroendocrine dysregulation, as well as social, cognitive and lifestyle adaptation to the present disease play a crucial role in disrupting sleep homeostasis and perceived wellness.

Disease-related lifestyle changes such as a sedentary life, medical treatment, disease-related pain and psychological factors such as self-isolation, a lack of social support and possible comorbidities with mental disorders might indirectly trigger the onset of sleep disturbances and impair well-being both in the short and the long term [52]. In fact, comorbid insomnia among patients suffering from chronic pain is frequently reported and is highly associated with anxiety and mood disorders which, in turn, worsen the general clinical picture [53,54]. Social and cognitive disease-related factors such as self-isolation, the social stigma of disease and scarce relationship networks can represent chronic stressors [55], leading to allostatic load in the long term, which is a condition characterised by the dysregulation of homeostatic processes [56]. In addition, the inflammatory biomarker cascade conveyed by the pathology itself can directly disrupt sleep quality and impair one’s perception of well-being. A detailed discussion is provided below.

### 4.1. Insomnia and Current Pathologies

Both experimental models of sleep deprivation and clinical studies on patients affected by chronic insomnia agree that a higher expression of C-reactive protein and a shift towards T helper type 2 cell activity during sleep disruption lead to an increased expression of interleukin-6, interleukin-10 and tumour necrosis factor, which is similar to the inflammatory activity associated with chronic diseases. The aforementioned molecular alterations have particularly been associated with severe insomnia [57,58,59]; this finding is consistent with the results of the present study, which shows that the probability of having severe insomnia in the general population is twice as high as having moderate insomnia among subjects exhibiting at least one pathology in the present.

Also, sleep disturbance has been reported to induce the persistent activation of the HPA axis, whose chronic activation, in turn, can induce glucocorticoid resistance in immune cells [60,61]. Furthermore, elevated levels of glucocorticoids, such as cortisol, have always been reported to be associated with higher blood pressure and high cholesterol, thus representing key components for cardiovascular risk [62].

In the present study, among the various heart conditions, only non-ischemic heart diseases such as heart valve disease, atrial fibrillation and arrhythmias were significantly correlated with the prevalence of insomnia; however, the pathogenesis of these conditions is inflammatory in nature as well, and is expressed in particular by an altered neutrophil to lymphocyte ratio and high levels of C-reactive protein, which are responsible for the comorbidities associated with hypertension [63,64,65]. Therefore, an opposite causative relationship between some heart diseases and insomnia can be assumed, such that alterations in glucocorticoids in hypertension, high cholesterol and heart conditions might severely alter sleep, as shown in the present study. The associations between these conditions and insomnia are corroborated by endocrinological studies reporting that growth hormone therapy is increasingly being used in heart disease treatment [66], as it enhances blood flow and some neurotrophic factors [67] by inducing T cell production [68,69]; this suggests that there is a functional deficiency of this hormone in heart diseases.

Growth hormone has a physiological peak that occurs after the onset of sleep during the first cycle of NREM sleep, which has been found to not occur during sleep deprivation conditions [70]. As studies have indicated its functional role in promoting slow-wave sleep [71], the lack of growth hormone in cardiovascular diseases and in some thyroid disorders [66] could be associated with sleep disruption via an alteration in the architecture of electrophysiological and biomolecular homeostasis during sleep. In particular, since the decreased expression of brain-derived neurotrophic factor has been associated with chronic insomnia [72], a lack of growth hormone alongside the reduced expression of slow-wave sleep might cause the inhibition of neurotrophic factor expression.

Additionally, reduced adenosine is reported to be correlated with impaired regulation of sleep homeostasis [57,73,74] and with high-cholesterol conditions, atherosclerosis, and other cardiovascular diseases [75,76]. The restoration of normal levels is shown to improve cholesterol homeostasis and anti-platelet activity [76]. In this line, chronic low levels of adenosine in high cholesterol, hypertension and heart diseases may lay the foundations for the exacerbation of insomnia too, as shown by the results of the present study.

Also, obstructive sleep apnoea and inflammatory skin disorders, such as psoriasis, pemphigus, urticaria and atopic dermatitis, share a common inflammatory substrate with high levels of interleukin-6 and tumour necrosis factor, likely paving the way for the onset of sleep disturbances [39,77,78,79,80]. In particular, because psoriasis has been classified as a psychosomatic disorder and, as such, often occurs with anxiety and depression concomitantly [81,82], these disorders might act as mediating variables that concur to favour insomnia.

### 4.2. Well-Being and Current Pathologies

The aforementioned conditions were also associated with impaired well-being. Indeed, the sickness response and HPA axis hyperactivation are also reported to be associated with chronic stress, thus impairing well-being and one’s quality of life [83]. In particular, high levels of cholesterol and C-reactive protein have been shown to display a positive correlation with low levels of well-being [84,85], thus explaining the relationship between heart and inflammatory diseases and impaired well-being. Additionally, the possible mediation of sociomedical sequelae in chronic diseases such as social stigma and the inability to pursue normal daily life activities due to disease might be involved in a worsening of well-being [86].

### 4.3. Insomnia, Well-Being and Current Psychiatric Disorders: The Vicious Circle That Could Prevent Resolution

Well-being was also shown to be correlated with the insomnia prevalence, suggesting that these two conditions are exacerbated by each other in a vicious cycle, further aggravating inflammatory activity and thus increasing the risk of the pathogenesis of other medical conditions [87].

In the present study, psychiatric disorders had the strongest association with insomnia and a low level of well-being compared to the other disorders. In the administered questionnaire, psychiatric disorders were not divided into specific categories. However, the most frequent psychiatric disorders among the Italian population are depression and anxiety [88], which are associated with high inflammatory activity. Indeed, mood disorders are characterised by a vigorous sickness response, HPA axis hyperactivation, the hyperactivation of the sympathetic system and the hypo-expression of serotonin, noradrenaline and dopamine, thus affecting the architecture of sleep via endocrinological and neurobiological pathways [15,89,90,91] and impairing well-being due to self-isolation and social stigma [92].

Although the results reporting a strict relationship between insomnia, impaired well-being and the current presence of pathologies can be discussed in line with the many studies highlighting these associations in specific patient populations [12,13,93,94], the pathophysiological basis underlying the association between resolved pathologies and the persistence of insomnia and impaired well-being has not yet been investigated in depth.

### 4.4. The Presence of Insomnia after Resolution of Pathologies Might Be Sustained by Inflammatory Track

In the present study, most resolved pathologies were correlated with impaired well-being, whereas only non-ischemic heart diseases and psychiatric disorders were associated with insomnia, even though they had already been resolved in the past. This can be explained by the fact that well-being is multifactorial, comprising psychological, social, emotional, and physical health aspects, and the impairment of one or more of them can directly affect the perception of well-being; meanwhile, the architecture of sleep could be more robust in order to support fundamental homeostatic functions.

Concerning the reason for the association between resolved pathologies and worsened well-being and insomnia, it is worth noting that most of the past pathologies that were investigated in the survey are chronic; therefore, subjects might have reported a resolution because the pathology was controlled by drugs. However, some studies have reported that several pathologies cause persistent inflammation once resolved, reporting that C-reactive protein is the prototypical marker of residual inflammation [95]. In particular, residual systemic inflammation with chronically elevated C-reactive protein levels has been reported in several chronic inflammatory conditions such as obesity, diabetes, myocardial infarction, high cholesterol [96], depression, hypothyroidism [97], rheumatoid arthritis [98], autoimmune diseases, and cancer [99]. For this reason, C-reactive protein has been suggested as a prognostic factor for cardiovascular risk [100]. Consistent with the results of the present study, non-ischemic heart disease was one of the resolved pathologies that positively correlated with current insomnia. In fact, as mentioned above, one of the main inflammatory biomarkers that is highly expressed in both acute and chronic sleep disturbances is C-reactive protein [14,57,101], suggesting the pivotal role of this biomarker in the exacerbation of insomnia induced by past medical conditions that may have left a sort of inflammatory track. Further evidence regarding the role of long-term immune dysregulation is given by recent studies on “Long-COVID syndrome”, which comprises neurological, pulmonary, cardiovascular, endothelial, and gastrointestinal complications [102,103].

The interplay between persistent inflammation and sleep alterations may be multifactorial, depending also on the disruption of circadian rhythms, including light hygiene [104]; this, in turn, has a strong association with cardiometabolic disorders [105,106], neuroinflammation [107], and neurodegenerative disorders [108], thus contributing to the effect had by C-reactive protein in a bidirectional relationship [109].

### 4.5. Well-Being and Resolved Pathologies

Although impaired well-being was also associated with high plasma levels of C-reactive protein [84,85], multiple factors might have contributed to its impairment. Indeed, pathology-induced lifestyle changes, non-adherence to follow-up treatment [110], a lack of a relationship network, the perception of illness and dysfunctional coping strategies [111] are crucial mediating factors that could have already been present during the pathology course and could still have an impact in the present, sustaining a persistent impairment of well-being.

However, since high levels of C-reactive protein have been found in subclinical conditions as well [99], pathologies that were declared to be resolved in the present study might have not completely healed, might have evolved into a subclinical condition, or might be controlled by medication, thus still having a strong impact on sleep quality and well-being. For instance, sleep has been reported to be disrupted in patients affected by ulcerative colitis even when symptoms were no longer active [87].

### 4.6. Relationship between Insomnia, Well-Being and Resolved Psychiatric Disorders

Interestingly, in the present study, psychiatric disorders were the only ones to have a strongly significant association with insomnia and a low level of well-being, both in the present and when resolved. Mood disorders, insomnia and well-being are intimately intertwined, and their relationship is determined by multiple factors: first, C-reactive protein is the most promising inflammatory index for the prognosis of depression, thus suggesting possible relapses or the onset of other medical conditions [112,113]. Additionally, cognitive and emotional factors related to depression and anxiety such as hyperarousal, rumination, and experiencing negative emotions might contribute to the exacerbation of insomnia. Insomnia, in turn, can be exacerbated by maintenance factors such as sleep-related negative thoughts and an anxious state that culminate in a vicious cycle with increasingly impaired daytime functioning and social and occupational deficits, thus leading to the further worsening of insomnia and the exacerbation of inflammation [114,115].

## 5. Limitations and Future Perspectives

The present study suffers from some limitations:Well-being was assessed using WHO-5, which does not include the evaluation of the socio-relational sphere. Future investigations should also focus on these aspects.The assessment of the subjects’ cognitive and psychological characteristics such as coping strategies, personality traits and illness behaviour, and the assessment of medical aspects such as adherence to treatment and treatment side effects, are strongly recommended, since they might act as mediating variables in determining impaired well-being and the development of chronic insomnia.Psychological disorders were not well defined in the survey: although the most frequent psychological disorders are depression and anxiety, the ability to speculate about the associations between insomnia and well-being and specific psychological conditions was precluded, since psychological disorders were not divided into specific categories.Medical conditions were not defined according to the ICD-10.Most of the past pathologies that were investigated in the survey are chronic; therefore, subjects might have reported a resolution because the pathology was controlled by drugs.This work does not investigate the effects of chronic diseases on sleep quality and well-being; however, the relevance of the phenomenon should be highlighted: even very specific conditions (e.g., retinal degeneration [116]) can have a global impact on health.

## 6. Conclusions

The present study provided evidence of the complex relationship between pathologies relying on an inflammatory substrate, insomnia, and perceived well-being. In particular, the main results suggest that resolved medical conditions produce relevant socio-medical sequelae that continue to affect sleep quality and well-being, thus indicating how health can be impaired regardless of disease resolution.

In conclusion, this study could represent a step forward in overcoming the dichotomy between organic and mental disorders, thus promoting integrated and multifactorial interventions that could prevent the relapse of disease and individuals’ predisposition to comorbidities, highlighting systemic inflammation as a common pathognomonic substrate. From this perspective, the moment-by-moment quantification of individual behaviours (e.g., actigraphic measures of circadian rhythms [117] or data from personal digital devices [118]) could be an effective way to ensure the early diagnosis and monitoring of several medical conditions.

## Figures and Tables

**Figure 1 ijerph-21-00129-f001:**
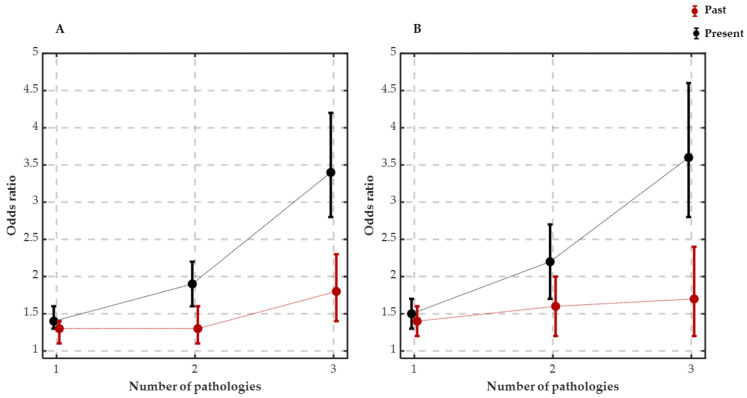
(**A**) Odds ratios for low well-being as a function of the number of current and resolved pathologies. Error bars show the odds ratio 95% confidence intervals for the presence of one, two and three pathologies for both the present and past. (**B**) Odds ratios for insomnia as a function of the number of resolved and current pathologies. Error bars show the odds ratio 95% confidence intervals for the presence of one, two and three pathologies for both the present and past.

**Figure 2 ijerph-21-00129-f002:**
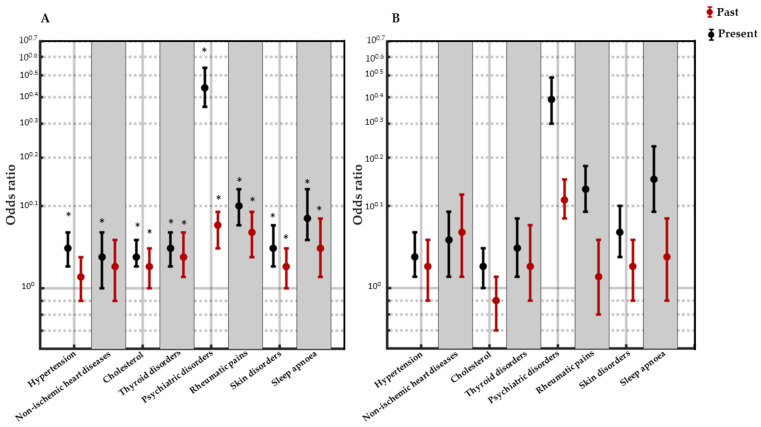
(**A**) Well-being as a function of current or past but resolved specific pathologies (odds ratios). Error bars show the 95% confidence interval. (**B**) Insomnia as a function of current or past but resolved specific pathologies (odds ratios). Error bars show the 95% confidence interval. * indicates pathologies for which the odds ratio from the logistic regression is statistically significant (*p* < 0.01). Y axes are expressed in log scale (10^0.1^ = 1.26; 10^0.7^ = 5.01).

**Table 1 ijerph-21-00129-t001:** Participants’ demographic characteristics.

Sample Characteristics	N (%)	
Gender	Males	Females
	2685 (48.3)	2876 (51.7)
Age, years	Males	Females
<20	122 (4.7)	122 (4.4)
20–29	361 (13.9)	503 (18.0)
30–39	363 (13.9)	487 (1.4)
40–49	499 (19.2)	576 (20.6)
50–59	601 (23.1)	574 (20.5)
60–69	500 (19.2)	407 (14.6)
>70	159 (6.1)	126 (4.5)
Marital status		
Unmarried	2145 (39.1)	
Married	2828 (51.5)	
Separate	177 (3.2)	
Divorced	219 (4.0)	
Widower	199 (2.2)	
Education		
None	7 (0.1)	
Primary Education	82 (1.5)	
Secondary Education	615 (11.2)	
Tertiary Education	366 (6.1)	
High School	2189 (39.8)	
Bachelor’s Degree	443 (8.1)	
Master’s Degree	1358 (24.7)	
Post-Graduation	470 (8.5)	
Employed		
No	2114 (38.4)	
Yes	3392 (61.6)	

**Table 2 ijerph-21-00129-t002:** Number of participants, stratified by gender, as a function of the WHO-5 and ISI scores and present and past pathologies.

	N(%)	
WHO-5	Males	Females
>50	1642 (61.2)	1503 (52.3)
≤50	1043 (38.8)	1373 (47.7)
ISI	Males	Females
<10	2168 (80.7)	2137 (74.3)
≥10	517 (19.3)	739 (25.7)
Present pathologies	Males	Females
No	205 (8.8)	228 (9.4)
One	295 (12.7)	295 (12.1)
Two	604 (25.9)	596 (24.5)
Three or more	1225 (52.6)	1317 (54.1)
Past pathologies	Males	Females
No	90 (3.9)	93 (3.8)
One	221 (9.5)	222 (9.1)
Two	521 (22.4)	684 (28.1)
Three or more	1497 (64.3)	1437 (59.0)

ISI, Insomnia Severity Index; WHO-5, World Health Organization well-being scale-5.

**Table 3 ijerph-21-00129-t003:** Mean scores for insomnia and well-being in participants currently displaying no pathologies or at least one pathology.

		N	Mean	S.D.
WHO-5	present pathologies	2223	48.66	21.06
	no present pathologies	2542	55.87	19.29
ISI	present pathologies	2223	6.92	4.92
	no present pathologies	2542	5.49	4.13

ISI, Insomnia Severity Index; S.D., standard deviation; WHO-5, World Health Organization well-being scale-5.

**Table 4 ijerph-21-00129-t004:** Mean scores for insomnia and well-being in participants who displayed no pathologies or at least one pathology in the past.

		N	Mean	S.D.
WHO-5	past pathologies	1831	50.10	20.43
	no past pathologies	2934	54.01	20.32
ISI	past pathologies	1831	6.80	4.71
	no past pathologies	2934	5.75	4.43

ISI, Insomnia Severity Index; S.D., standard deviation; WHO-5, World Health Organization well-being scale-5.

**Table 5 ijerph-21-00129-t005:** Binary logistic regression model of the association between a low level of well-being (WHO-5 < 50) and the presence of at least one current or resolved pathology. Reference category is “no pathologies”.

	Regression Weights		
Variables (d.f.)	O.R.	95% CI	*p* Value
Age (1)	0.99	0.98–1.00	0.43
Gender (1)	1.18	0.88–1.58	0.24
Age × gender (1)	1.00	0.99–1.00	0.44
Only present (1)	1.75	1.58–1.94	<0.0001
Only past (1)	1.33	1.21–1.47	<0.0001

CI, confidence intervals; d.f., degrees of freedom; (1), one degree of freedom.

**Table 6 ijerph-21-00129-t006:** Binary logistic regression model of association between insomnia (ISI > 10) and the presence of at least one current or resolved pathology. Reference category is “no pathologies”.

	Regression Weights		
Variables (d.f.)	O.R.	95% CI	*p* Value
Age (1)	0.98	0.97–0.99	0.02
Gender (1)	0.84	0.55–1.29	0.43
Age × gender (1)	1.01	1.00–1.02	0.01
Only present (1)	1.90	1.63–2.20	<0.0001
Only past (1)	1.43	1.24–1.60	<0.0001

CI, confidence intervals; d.f., degrees of freedom; (1), one degree of freedom.

## Data Availability

All data utilized for the analysis are available in the Appendix A. The data presented in this study are available in Appendix A, tipe of file: excell.

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
