# Peer review of "Impaired Well-Being and Insomnia as Residuals of Resolved Medical Conditions: Survey in the Italian Population"

_ijerph, 2024, doi:10.3390/ijerph21020129_

Round 1

Reviewer 1 Report

Comments and Suggestions for Authors

Thank you for this study on the association between past vs. present pathologies and association with insomnia symptoms as assessed by the ISI and wellbeing as assessed by the WHO-5. Why this is an interesting research question, the manuscript does not present the data in a way that is fully transparent and which bolsters the reader’s confidence in the validity of the results/conclusions. A discussion of study limitation is also sorely needed.

MAJOR COMMENTS

1.     Survey target population and response rate should be included in the Abstract and in the Methods. The representativeness of the resulting analysis sample/s should also be described in the Methods/Results.

2.     Given that the presence of medical conditions is a key variable in the study, all conditions evaluated should be listed in the Methods under section 2.1. It is not sufficient to copy and paste the questionnaire into the Supplementary Materials.

3.     There is reason to believe that insomnia differs by sex/gender, therefore all analyses should be stratified by sex/gender.

4.     Table 1 should include ISI, WHO-5, and number of pathologies for men and women. The function of Table 1 is to thoroughly describe the data.

5.     I would like to see a new table or figure which describes the % of participants with no/present/past disease and the mean outcome scores (ISI and WHO-5) in these groups.

6.     Results for binary logistic regression in Tables 2 and 3 should be consistently presented to 2-decimal places – currently, there is inconsistent rounding of the OR and 95% CI. The table is also currently unclear which is considered the referent category for categorical variables. This need to be made explicit in the table.

7.     There are 95% CI missing from Table 3.

8.     The modelling does not take into account the sociodemographic characteristics in Table 1, apart from age and gender. The authors need to justify why this method was chosen.

9.     Figure 2 is a good way to summarise the findings, however, the authors should present the y-axis on a logarithmic scale, rather than linear scale. Only on a log scale can the size of the effects/error bars be meaningfully compared.  

10.  Limitations of the study need to be Discussed.

MINOR COMMENTS

1.     Key numeric findings should be described in the Abstract.

Comments on the Quality of English Language

N/A

Author Response

We thank you for the stimulating comments and concerns, which we have taken in the utmost consideration and which we believe have led to improvements.
In the following we report point by point the issues you raised along with our response. We answered each question, and we modified the manuscript accordingly.

REVIEWER 1

Thank you for this study on the association between past vs. present pathologies and association with insomnia symptoms as assessed by the ISI and wellbeing as assessed by the WHO-5. Why this is an interesting research question, the manuscript does not present the data in a way that is fully transparent and which bolsters the reader’s confidence in the validity of the results/conclusions. A discussion of study limitation is also sorely needed.

MAJOR COMMENTS

  1. Survey target population and response rate should be included in the Abstract and in the Methods. The representativeness of the resulting analysis sample/s should also be described in the Methods/Results.

Authors: The number of participants who fully answered to both WHO and ISI was 5561, namely 53.13% of the recruited subjects.

Regarding study representativeness, this research uses individual level data from the IPSAD study, a cross-sectional survey conducted by the Italian National Research Council (CNR-IFC). IPSAD is the only Italian survey providing nationally representative data on substance use and risk behaviours in the population aged 15–74. Data are collected through postal self-administered and anonymous paper-and-pencil questionnaires from a proportional stratified randomised sample. A detailed description of the survey methodology is provided elsewhere (DPA, 2014).

DPA - Dipartimento Politiche Antidroga Relazione annuale al Parlamento, Technical report

Presidenza del Consiglio dei Ministri (2014)

https://www.politicheantidroga.gov.it/media/2987/relazione-annuale-al-parlamento-2015-dati-2014.pdf

  1. Given that the presence of medical conditions is a key variable in the study, all conditions evaluated should be listed in the Methods under section 2.1. It is not sufficient to copy and paste the questionnaire into the Supplementary Materials.

Authors: We specified all the conditions investigated in the Methods, as suggested.

  1. There is reason to believe that insomnia differs by sex/gender, therefore all analyses should be stratified by sex/gender.

Authors: We thank the reviewer for his comment. Our results also reveal a role of age and of age x gender interaction in insomnia. In fact, we took into account these variables in our analyses by including them in the regression models. Significant effects of age and age x gender interaction related to the ISI binary model are indicated in new Table 6. In addition, significant effects of these variables related to the multinomial models are shown in Table S6 and Table S8 of Supplementary Materials. We also specified in the “Results” section significant effects of age and gender and age x gender interaction for binary models related to each pathology and for multinomial models.

We added two contingency tables showing associations between gender and insomnia and gender and well-being, respectively (Table S1 and Table S2).

  1. Table 1 should include ISI, WHO-5, and number of pathologies for men and women. The function of Table 1 is to thoroughly describe the data.

Authors: According to Reviewer’s suggestion, we added a new table (Table 2) showing the number of men and women reporting ISI and WHO-5 scores below or above cut-off and the number of participants displaying one, two, three or no current and past pathologies..

Please note that all count tables are now based on the number of subjects who completed both ISI and WHO-5 questionnaires. Data that were edited in Table 1 are highlighted in yellow.

  1. I would like to see a new table or figure which describes the % of participants with no/present/past disease and the mean outcome scores (ISI and WHO-5) in these groups.

Authors: We added two new tables (Table 3 and Table 4) showing the number of participants who displayed no pathologies or at least one pathology in the present or in the past and the related well-being and insomnia scores.

  1. Results for binary logistic regression in Tables 2 and 3 should be consistently presented to 2-decimal places – currently, there is inconsistent rounding of the OR and 95% CI. The table is also currently unclear which is considered the referent category for categorical variables. This need to be made explicit in the table.

Authors: We thank the reviewer for his comment. We uniformed the number notation and added a note for the reference category in the regression.

  1. There are 95% CI missing from Table 3.

We apologize for the typos. In the revised Table 3, all 95%CI has been correctly reported.

  1. The modelling does not take into account the sociodemographic characteristics in Table 1, apart from age and gender. The authors need to justify why this method was chosen.

We appreciate the reviewer’s comment for bringing this issue to our attention. The variables in Table 1 just provide a summary description of the sample. On the other hand, the survey makes available hundreds of descriptive variables of subjects' characteristics and habits, but in the absence of specific assumptions about them, it would have been hard to remove a potential influence of each of them. Also, many of these variables could show collinearity (e.g., between age and employment status) making model estimation more complex. We make explicit this choice in the Methods section.

  1. Figure 2 is a good way to summarise the findings, however, the authors should present the y-axis on a logarithmic scale, rather than linear scale. Only on a log scale can the size of the effects/error bars be meaningfully compared.  

Authors: We thank the reviewer for the suggestion. The revised figures have been modified accordingly.

  1. Limitations of the study need to be Discussed.

Authors: We welcome the suggestion by highlighting limitations in a specific section before Conclusions.

MINOR COMMENTS

  1. Key numeric findings should be described in the Abstract.

Authors: Most relevant odds ratios have now been reported in the Abstract.

Reviewer 2 Report

Comments and Suggestions for Authors

The research is very interesting. However, I have few questions below

1. What is the definition of having sleep disorders during pregnancy? It was clear on the manuscript.

2. On the abstract, please specify the accuracy (sensitivity, specificity) for the validation set in the same way as the training set.

3. Table 3 and figure 3 are quite difficult to understand. Can the authors explain more clearly how to use it?

Comments on the Quality of English Language

The quality of English is fine.

Author Response

REVIEWER 2

The research is very interesting. However, I have few questions below

  1. What is the definition of having sleep disorders during pregnancy? It was clear on the manuscript.

We thank the reviewer for the suggestion. However, the present paper focused on the past and present diseases. Authors could consider the pregnancy condition for a future work.

  1. On the abstract, please specify the accuracy (sensitivity, specificity) for the validation set in the same way as the training set.

Authors: In the present study traditional logistic regression was implemented instead of a machine learning-based approach. This could be used as a suggestion for future studies.

  1. Table 3 and figure 3 are quite difficult to understand. Can the authors explain more clearly how to use it?

Authors: We suppose it is Figure 2. We modified it by converting the y axis in log scale instead of linear scale for a better comprehension.

We apologize for the typos in Table 3. In the revised Table 3, the error regarding confidence intervals has been corrected.

Reviewer 3 Report

Comments and Suggestions for Authors

In this paper the authors report connections between number of current resolved medical conditions with increasingly compromised sleep quality and well-being. A paper is informative, well-written and should be interesting to a wide readership.

I can recommend it for publication.

However, there are several comments that may help to improve discussion of the results:

  1. It is not clear, what classification was used to choose and define medical conditions in this work, that differs from ICD-10, which is usually used, as in UK Biobank (e.g.,doi: 10.1016/S2666-7568(23)00056-9; doi: 10.1080/07420528.2018.1454458).

  2. While discussing the presence of insomnia after resolution of pathologies that can be sustained by inflammatory track, the authors paid attention to C-reactive protein as a suggested prognostic factor for cardiovascular risk related to insomnia, related to “systemic residual inflammation”. Of note, the interplay between persistent low-grade inflammation and circadian disorders including impaired sleep can be multi-factorial and above all, rely on circadian factors of reduced sleep quality and circadian disruption: low 24-h activity / circadian amplitude (e.g.,https://doi.org/10.3390/app12189220), low daylight, and /or light at night (doi: 10.1126/science. adg5277; doi: 10.1016/j.heliyon.2023.e17837; doi: 10.1007/s00281-021-00899-0). These factors have a strong association with cardiometabolic disorders (e.g., doi: 10.1073/pnas.1516953113; doi: 10.3390/ijms241814145) but also neuroinflammation / neurodegeneration (doi: 10.1186/s40035-023-00340-6), affecting also C-reactive protein (doi: 10.1073/pnas.1516953113).

  3. The authors did not include metabolic (endocrine (not only “thyroid disorders”), diabetes) (doi: 10.1093/sleep/zsac130; doi: 10.1001/jamainternmed.2019.0571) and neurodegenerative (neirologic) (doi: 10.1016/j.jsmc.2017.09.006; doi: 10.4103/1673-5374.332149) morbidity in their work, however, these pathologies are closely linked with impaired sleep and well-being and depend on light hygiene that can either enhance or compromise sleep and well-being (e.g., doi: 10.1111/jsr.13471).

  4. Notably, not just resolving but, vice versa, progression of pathology, for example neurodegenerative retinal damage affecting light perception, is coupled with worsening of several conditions of health and well-being, including sleep (doi: 10.1080/07420528.2019.1566741), mood (doi: 10.1016/j.jad.2023.04.039), and cholesterol metabolism (https://doi.org/10.3390/app12136374).

  5. The authors may wish to consider in discussion that, as shown in very recent publications, applying objective, i.e., actigraphic measures of sleep and circadian rhythm may facilitate early diagnostics of numerous pathologies (doi: 10.1016/S2666-7568(23)00056-9.) and frailty risks (doi: 10.1038/s41467-023-42727-z.).

Author Response

REVIEWER 3

In this paper the authors report connections between number of current resolved medical conditions with increasingly compromised sleep quality and well-being. A paper is informative, well-written and should be interesting to a wide readership.

I can recommend it for publication.

However, there are several comments that may help to improve discussion of the results:

  1. It is not clear, what classification was used to choose and define medical conditions in this work, that differs from ICD-10, which is usually used, as in UK Biobank (e.g.,doi: 10.1016/S2666-7568(23)00056-9; doi: 10.1080/07420528.2018.1454458).

Authors: We thank the Reviewer for highlighting a limitation of this study that we have declared in the revised version (Limitations section). Medical conditions in the survey were investigated mainly based on the expected prevalence in the general population, without the sake of being complete and this choice has precluded the possibility to cluster them according to ICD-10.

  1. While discussing the presence of insomnia after resolution of pathologies that can be sustained by inflammatory track, the authors paid attention to C-reactive protein as a suggested prognostic factor for cardiovascular risk related to insomnia, related to “systemic residual inflammation”. Of note, the interplay between persistent low-grade inflammation and circadian disorders including impaired sleep can be multi-factorial and above all, rely on circadian factors of reduced sleep quality and circadian disruption: low 24-h activity / circadian amplitude (e.g.,https://doi.org/10.3390/app12189220), low daylight, and /or light at night (doi: 10.1126/science. adg5277; doi: 10.1016/j.heliyon.2023.e17837; doi: 10.1007/s00281-021-00899-0). These factors have a strong association with cardiometabolic disorders (e.g., doi: 10.1073/pnas.1516953113; doi: 10.3390/ijms241814145) but also neuroinflammation / neurodegeneration (doi: 10.1186/s40035-023-00340-6), affecting also C-reactive protein (doi: 10.1073/pnas.1516953113).

Authors: We thank the Reviewer. In the revised Discussion, we have emphasized the multifactorial nature of interaction between insomnia and past pathologies.

  1. The authors did not include metabolic (endocrine (not only “thyroid disorders”), diabetes) (doi: 10.1093/sleep/zsac130; doi: 10.1001/jamainternmed.2019.0571) and neurodegenerative (neirologic) (doi: 10.1016/j.jsmc.2017.09.006; doi: 10.4103/1673-5374.332149) morbidity in their work, however, these pathologies are closely linked with impaired sleep and well-being and depend on light hygiene that can either enhance or compromise sleep and well-being (e.g., doi: 10.1111/jsr.13471).

Authors: We highlighted the role of circadian rhythm in cardiometabolic and neurodegenerative disorders and the bidirectional relationship between them.

  1. Notably, not just resolving but, vice versa, progression of pathology, for example neurodegenerative retinal damage affecting light perception, is coupled with worsening of several conditions of health and well-being, including sleep (doi: 10.1080/07420528.2019.1566741), mood (doi: 10.1016/j.jad.2023.04.039), and cholesterol metabolism (https://doi.org/10.3390/app12136374).

We thank the Reviewers for highlighting the impact of chronic diseases on sleep and wellbeing.

  1. The authors may wish to consider in discussion that, as shown in very recent publications, applying objective, i.e., actigraphic measures of sleep and circadian rhythm may facilitate early diagnostics of numerous pathologies (doi: 10.1016/S2666-7568(23)00056-9.) and frailty risks (doi: 10.1038/s41467-023-42727-z.).

We very gladly take up the Reviewer's suggestion, we have added this consideration in the Conclusions section.
